# Evidence of Flavonoids on Disease Prevention

**DOI:** 10.3390/antiox12020527

**Published:** 2023-02-20

**Authors:** Meng Li, Mengqi Qian, Qian Jiang, Bie Tan, Yulong Yin, Xinyan Han

**Affiliations:** 1Hainan Institute, Zhejiang University, Sanya 572000, China; 2College of Animal Sciences, Zhejiang University, Hangzhou 310058, China; 3College of Animal Science and Technology, Hunan Agricultural University, Changsha 410128, China

**Keywords:** flavonoids, antioxidants, obesity, cancer, neurodegenerative diseases

## Abstract

A growing body of evidence highlights the properties of flavonoids in natural foods for disease prevention. Due to their antioxidative, anti-inflammatory, and anti-carcinogenic activities, flavonoids have been revealed to benefit skeletal muscle, liver, pancreas, adipocytes, and neural cells. In this review, we introduced the basic classification, natural sources, and biochemical properties of flavonoids, then summarize the experimental results and underlying molecular mechanisms concerning the effects of flavonoid consumption on obesity, cancers, and neurogenerative diseases that greatly threaten public health. Especially, the dosage and duration of flavonoids intervening in these diseases are discussed, which might guide healthy dietary habits for people of different physical status.

## 1. Introduction

With the penetration of food and nutrition science in medical research, the significance of functional plant foods for disease prevention has been highlighted [1]. Flavonoids, the bioactive polyphenolic phytochemicals existing in natural foods, receive a wide-range of attention due to their anti-inflammatory and antioxidative properties and exhibit inspiring outcomes in various experimental and clinical studies [2,3,4,5]. Generally, flavonoids are a 15-carbon (C6–C3–C6) skeleton that is composed of two benzene rings (C6) and 3-carbon linking chain (C3) [6]. Depending on the chemical structure, flavonoids are divided into flavonols, flavan-3-ols, anthocyanidins, isoflavones, flavanones, and flavones. In the current review, we will introduce the basic classification, natural sources, and biochemical properties of 21 typical flavonoids, the classification and chemical structures of which are shown in the Figure 1, then summarize their experimental results and underlying molecular mechanisms concerning the effects on obesity, cancers, and neurogenerative diseases that greatly threaten public health.

Chronic inflammation and oxidative stress are involved in the pathogenesis of obesity, cancer, and neurodegenerative diseases. Flavonoids are emerging as a potential therapeutic molecule for these diseases due to their anti-inflammatory and antioxidative properties [7,8,9]. For instance, apigenin (a flavone derived from chamomile tea and celery) was reported to have strong anti-adipogenesis activity preventing obesity [10]. Genistein, the predominant soy isoflavone, has also shown its antioxidative and neuroprotective activities in various experimental models [11,12]. There is evidence indicating that plasma isoflavone concentration was inversely associated with the risk of breast cancer among Chinese women and prostate cancer in Asian men [13,14]. Likewise, various flavonoids have been revealed to affect skeletal muscle, liver, pancreas, adipocytes, immune cells, and brain neural cells via different mechanisms that are relevant to diseases development [2,15,16,17,18]. The potential mechanisms of flavonoids intervening in obesity, cancers, and neurogenerative diseases are briefly summarized in Figure 2. Previous reviews focus on the effects and mechanisms of flavonoids on diseases. Here, we will review and discuss the innovative findings in the field of flavonoid research, including the effects and molecular mechanisms of different flavonoids on obesity, cancer, and neurodegenerative disease prevention, especially, the discussion of outcomes with significant differences in dosage and duration of flavonoids treatment.

## 2. The Effects of Flavonoids on Obesity

Obesity is a prevalent disease that is influenced by complicated factors, such as dietary patterns, genetic factors, and behavioral activity [19]. The incidence of obesity is greatly increasing worldwide. Currently, more than 1.9 billion people are suffering from obesity and its concurrent diseases, such as diabetes and cardiovascular diseases [20]. The World Health Organization (WHO) defines obesity as an abnormal or excessive accumulation of fat that can affect health. In terms of pathogenesis, excessive energy is stored inside the adipocytes in the form of triglycerides, during which these adipocytes increase in volume (hypertrophy), leading to obesity, accompanied by chronic inflammation and insulin resistance [21,22]. Several studies have demonstrated that dietary patterns rich in flavonoids are associated with healthy body weight and metabolism [1,19,22,23,24]. Besides, the roles of flavonoids in anti-adipogenesis, energy expenditure, and insulin resistance properties have been clarified in different experimental models (Table 1). Below, we summarized the evidence of different flavonoids intervening in obesity development, focusing on the aspects of effective dosage, duration, effects, and definite mechanisms.

### 2.1. Anti-Adipogenesis Effects

Adipose hyperplasia is the basal pathogenesis of obesity development. Adipogenesis undergoes two differentiation stages: one is the determination stage, during which multipotent mesenchymal stem cells (MSCs) differentiate into preadipocytes; another stage is terminal differentiation, during which preadipocytes acquire metabolic features of mature adipocytes [37,38]. Emerging evidence suggests that flavonoids regulate adipogenesis through different mechanisms. Peroxisome proliferator-activated receptor-gamma (PPARγ), abundantly expressed in adipose tissue, is involved in the anti-adipogenesis activity of flavonoids by regulating adipocyte differentiation [39]. For instance, black soybean-derived anthocyanins have been revealed to inhibit preadipocyte’s terminal differentiation by reducing PPARγ expression [25]. Fisetin can suppress the transcriptional activity of PPAR-γ via activating NAD (+)-dependent deacetylase SIRT1, thus inhibiting adipogenesis [26,40,41]. Besides, there is evidence that flavonoids (e.g., isorhamnetin, genistein, and daidzein) regulate the differentiation potential of preadipocytes into mature adipocytes via the wingless-type MMTV integration site family/canonical (Wnt/β-catenin) pathway [27,42]. In terms of the crosstalk among Wnt/β-catenin, C/EBP, and PPAR-γ, the binding of Wnt to cell surface receptors leads to β-catenin transferring into the nucleus, which then interacts with transcription factors in the nucleus and suppresses CCAAT/enhancer-binding protein family members (C/EBP) and PPAR-γ [43]. Other anti-adipogenic mechanisms of flavonoids include mitotic clonal expansion, cell-cycle arrest, activation of AMP-activated protein kinase (AMPK), reduction of C/EBPα-GLUT4 signaling, and inhibition of mTOR signaling [10,44,45]. Representatively, apigenin, a main flavone found in herbs, can arrest the cell cycle at G0/G1 phase through lowering cyclin D1 and CDK4 expression and decreasing adipogenic expression by activating the AMPK pathway [10,46].

### 2.2. Promotion of Energy Expenditure

In several obesity models, flavonoid supplementation prevented weight gain without any influences on energy intake or absorption [47,48], indicating potential effects on energy expenditure. White adipose tissue browning plays a key role in promoting energy expenditure and reducing adiposity [49]. Flavonoids have been revealed to activate the AMPK/PPAR-γ pathway in the sympathetic nervous system, thereby promoting browning and thermogenesis in brown and subcutaneous adipose tissues [50,51]. For example, in the high-fat-fed mice model, luteolin increased energy expenditure via activating AMPK/PGC-1α signaling and up-regulating gene expressions of thermogenesis [28]. Flavan-3-ols, a natural flavonoid that is abundant in chocolate, was revealed to stimulate the sympathetic nervous system and promote adrenalin release, which enhances AMPK and PGC-1α activation and energy expenditure [29]. Mitochondrial biogenesis is another crucial regulator for thermogenesis and energy expenditure in adipose and muscle tissues. Several studies showed that flavonoids can alleviate obesity via reversing the mitochondrial biogenesis suppression [52,53,54]. For example, epigallocatechin gallate (EGCG) decreased body weight by regulating mitochondria biogenesis in high-fat-fed obese mice. These actions of EGCG were mediated partially by AMPK activation and mitochondrial DNA replication in brown adipose tissues [30]. Furthermore, other anti-obesity mechanisms of EGCG and catechins from tea were attributed to the decreased food digestibility and enhanced fat oxidation [31,55].

### 2.3. Alleviation of Insulin Resistance

Insulin resistance is a pathological condition that causes metabolic abnormalities in obesity and diabetes. In obese individuals, adipose tissue release increased amounts of hormones, glycerol, pro-inflammatory cytokines, and non-esterified fatty acids, which are involved in insulin resistance [56]. In turn, insulin resistance can decrease glucose transport metabolism in skeletal muscle and adipocytes, and increase hepatic glucose production, thus contributing to obesity development [57]. When insulin signaling is defective, the insulin receptor substrate (IRS) phosphorylation and the PI3K/Akt pathway are inhibited [58,59], which damages the AMPK phosphorylation and GLUT4 translocation to the cellular membrane, contributing to decreased glucose uptake in the skeletal muscle and adipose tissue [34]. In the liver, the defective PI3K/Akt pathway promotes gluconeogenesis by increasing the expression of phosphoenolpyruvate carboxykinase (PEPCK) and glucose 6-phosphate (G6P) while inhibiting glycogen synthesis by suppressing glucokinase (GK) and glycogen synthesis kinase (GSK). Numerous experimental studies have revealed that flavonoids can greatly improve insulin sensitivity via glucose and insulin associated signaling pathways. For example, genistein can improve insulin signaling by restoring IRS phosphorylation and PI3K/Akt pathway in mice with a high-fat–high-fructose diet [32]. Myricetin, a flavonoid component in a variety of plants, has been revealed to improve insulin sensitivity by directly restoring IR/IRS1 phosphorylation, or indirectly binding µ-opioid receptor [33,34]. Restoring normal glucose uptake in the skeletal muscle and adipose tissue are involved in another mechanism of flavonoids improving insulin sensitivity. For instance, in high-fructose-fed and high-fat-fed mice, supplementation with grape seed-derived procyanidins (formed from catechin and epicatechin molecules) improves insulin sensitivity by increasing the expression of GK and hepatic glycogen concentration in mice and HepG2 cells [36]. In the liver of type-2 diabetic mice, it has been reported that naringin and hesperidin promote glycogen synthesis by increasing expression of GK, and inhibit gluconeogenesis by decreasing the mRNA level of PEPCK and G6P [35]. Besides, green tea extract (rich in epicatechin, epicatechin gallate, epigallocatechin, and epigallocatechin gallate) significantly improves insulin sensitivity through enhancing glucose transporters [36,60]. The potential mechanisms of how flavonoids intervene in insulin sensitivity are shown in Figure 3.

In summary, flavonoids inhibit adipogenesis through reducing PPAR-γ level, adipogenic gene expression and cell cycle, and increasing SIRT1 expression and Wnt/β-catenin signaling; Flavonoids promote energy expenditure by activating AMPK/PGC-1α signaling, stimulating sympathetic nervous system, promoting Mitochondrial DNA replication and AMPK activation, enhancing fat oxidation, and decreasing food digestibility; Flavonoids alleviate insulin resistance via increasing phosphorylation of IR and IRS1, expression of IRS-1, PI3K, Akt and GK, and hepatic glycogen concentration, directly binding to µ-opioid receptor, and decreasing PEPCK and G6P.

## 3. Anti-Carcinogenic Roles of Flavonoids

Cancer is a major public health problem and the most significant barrier to improving life expectancy worldwide. In 2020, an estimated 19.3 million new cancer cases and almost 10.0 million cancer deaths occurred worldwide [61]. Female breast cancer (11.7%) has surpassed lung cancer as the most incidence cancer, followed by lung (11.4%), colorectal (10.0%), prostate (7.3%), and stomach (5.6%) cancers [61]. According to GLOBACAN 2020, the global cancer burden is expected to be 28.4 million cases in 2040, a 47% rise from 2020 [61]. Therefore, the significance of cancer prevention has been highlighted. In numerous population-based studies, flavonoids have been demonstrated to exert anti-carcinogenic properties against different types of cancers [12,62,63,64]. For example, a study of predominately postmenopausal women reported an inverse association between dairy intake of flavanones and incidence of lung cancer in current and past smokers [65]. Besides, a prospective study of 42,099 Swedish women assessed the association between tea consumption and breast cancer risk and found that those who drank 1–2 cups tea/day had a higher risk of breast cancer [66]. Besides, numerous case-control and prospective cohort studies demonstrated that dietary flavonoids decreased the risk of lung, breast, prostate, stomach, liver, and colorectal cancers [65,67,68,69,70,71]. The mechanisms of anti-carcinogenic effects of flavonoids involve multiple cellular processes, including DNA damage, programmed cell death, inflammatory tumor microenvironment, and tumor angiogenesis.

**Table 2 antioxidants-12-00527-t002:** Anti-carcinogenic roles of flavonoids in the clinical study and experimental models.

Flavonoids	Dosage	Duration	Models	Effects	Molecular Mechanisms	Ref.
Quercetin	15 mg/day	-	Population-based, case-control study	↓Lung cancer	↓ CYP1A1	[72]
Quercetin	25 mg/day	-	A case-control study in Finland	↓Lung cancer	-	[73]
Flavonols	15 mg/day	-	A case-control study in Italian	↓Breast cancer	-	[74]
Naringenin	10–160 μM	24–72 h	Gastric cancer SGC7901 cell line	↑Apoptosis	↑Apoptotic proteins ↓AKT	[75]
Hesperetin	100–300 μM	24–48 h	Esophageal cancer cells	↑Apoptosis	↑ROS production ↑Intracellular caspase-9, caspase-3, Apaf-1	[76]
Quercetin	20–100 μM	24–72 h	BT-474 breast cancer cells	↑Caspase-dependent extrinsic apoptosis	↑Caspase-8 and caspase-3 ↓STAT3 signaling	[77]
Fisetin	40–120 μM	24–96 h	Prostate cancer cells	↑Autophagy	↓mTOR signaling pathway	[78]
Quercetin	30–90 μM	48 h	Human breast cancer cell line	↑Autophagy ↓mTOR activity	↓Proteasome	[79]
Apigenin	12.5–50 μM	24 h	HCT116 human colon cancer cells	↑Apoptosis	↓Autophagy	[80]
Apigenin	20–80 μM	24–48 h	Human breast cancer cells	↑Apoptosis	↓Autophagy	[81]
Delphinidin	120–180 μM	48 h	Human prostate cancer cells	↓Cell growth	↑Autophagy	[82]
Anthocyanin	30–150 μM	72 h	Human oral cancer cells	Anti-metastatic properties	↓Autophagy	[83]
Naringenin	100 mg/kg	72 h	Breast cancer resection model mice	↓Metastases outgrowth	↓Treg-induced immunosuppression	[84]
Methlut	1–100 μM	2–24 h	Mast cell	Inflammatory conditions	↓Intracellular calcium ↓NF-kB activation	[85]

Notes: CYP1A1, Cytochrome P450 Family 1 Subfamily A Member 1; AKT, protein kinase B; mTOR, mammalian target of rapamycin; NF-κB, nuclear factor kappa-light-chain-enhancer of activated B cells; ROS, reactive oxygen species; STAT3, signal transducer and activator of transcription 3. ↓, downregulate; ↑, upregulate.

### 3.1. Apoptosis Induction

Apoptosis is a type of programmed cell death in response to diverse signals of intracellular damage and is critical for tissue homeostasis [86]. There are two principal pathways involved in cell apoptosis (Figure 4). One intrinsic pathway is triggered by diverse intracellular stresses, which activate cytochrome c, Apaf-1, and caspase-9 to form an apoptosome through regulation of BH3-only and Bcl2-like proteins, thus promoting apoptosis [87,88]. The other is the extrinsic pathway induced when the TNF family binds to death receptors on the cell surface, which activate caspase-8 and caspase-3 through Fas-associated death domain protein (FADD) and provoke cellular destruction [88,89,90]. Impaired cell apoptosis is a central step towards cancer development [91]. Studies targeting apoptosis induction for cancer therapies are emerging, including regulation of apoptotic proteins, ROS production, and DNA damage [92,93]. Hesperetin was reported to induce apoptosis in esophageal cancer cells via ROS accumulation and mitochondrial-mediated intrinsic pathway [76]. Naringenin can induce apoptosis in gastric cancer SGC7901 cell lines and inhibit cell proliferation, migration, and invasion [75]. Quercetin is reported to promote apoptosis via increasing pro-apoptotic proteins and decreasing anti-apoptotic proteins in breast cancer cells [48,77]. When co-administered with EGCC or metformin, quercetin can induce apoptosis and decrease the viability, migration, and invasion of cancer cells [94,95]. Other flavonoids, including flavones, anthocyanidins, and isoflavonoids were also associated with anticancer activities through intrinsic and extrinsic pathways of apoptosis [96,97,98,99].

### 3.2. Regulation of Autophagy

In addition to apoptosis, numerous studies have revealed a significant effect of flavonoids on anticancer by modulating the autophagy network. Autophagy is a process where intracellular components degrade and recycle in response to cellular starvation or stress [100,101]. Autophagy plays a dual role in the development of cancer [102,103]. In the early stage of tumorigenesis, autophagy can suppress the proliferation of cancer cells, but at a later stage, it promotes the survival of cancer cells under oxygen-poor and low-nutrient conditions [103,104,105]. PI3K–AKT–mTOR signaling pathway is the main regulator of autophagy, where class Ⅰ PI3K inhibits autophagy through the mTORC1 pathway, and class Ⅲ PI3K acts as an initiator of autophagy [100,106]. In the early stage of cancer development, flavonoids can induce autophagy to facilitate apoptosis or promote the death of cancer cells that have a defective apoptotic pathway [107]. For instance, the bioflavonoid quercetin can inhibit proteasome activity in human cancer cells, thereby triggering macroautophagy and blocking mTOR activity in human cancer cells [79]. Fisetin, a natural flavonoid found in fruits and vegetables, was reported to induce autophagic-programmed cell death through inhibition of both mTORC1 and mTORC2 complexes in prostate cancer cells [78]. Besides, genistein and baicalein can also target the PI3K–AKT–mTOR pathway and stimulate autophagic cell death in various cancer cells [108,109]. At a later stage of cancer progression, autophagy functions as a survival mechanism in cancer cells against apoptosis. A study showed that the use of autophagy inhibitors can enhance anthocyanins-induced apoptosis in liver cancer cells [110]. Other studies also showed that autophagy inhibition promotes apigenin-induced apoptosis in human colon cancer cells and breast cancer cells [80,81]. Besides, fisetin was reported to exert cytotoxicity in human breast cancer MCF-7 cells by inhibition of autophagy and induction of caspase-7-associated apoptosis, without any detectable cytotoxicity in non-tumorigenic MCF-10A cells [111].

### 3.3. Targeting NF-κB

NF-κB is a crucial mediator linking immunity and inflammation to cancer development and progression [112]. NF-κB is activated by chronic inflammation or viral/microbial infections, and the activated NF-κB has different effects on different cancer cells. In epithelial cells, NF-κB activation has a negative impact on cancer development, but an opposite effect on inflammatory cells [113]. Bilal et al. showed that delphinidin treatment inhibits the growth of human prostate cancer cells by activating NF-κB without affecting normal human prostate epithelial cells [82]. However, another study demonstrated that anthocyanins from Black Rice exert antimetastatic properties by inhibiting NF-κB expressions in human oral cancer cells [83]. Therefore, targeting the initial cause of NF-κB activation (immune and inflammatory disorders) might be more effective measures to prevent cancer development. For example, cyanidin was reported to attenuate IL-17A–induced skin hyperplasia, alleviate airway hyperreactivity, and inhibit inflammation induced by IL-17–producing TH17 cells in severe asthma and steroid-resistant model [114]. The molecular basis of these actions is that cyanidin specifically recognizes the IL-17A binding site in IL-17A receptor subunit (IL-17RA) and inhibits the interaction of IL-17A/IL-17RA [114]. Tregs, which are significantly increased in cancer patients and play an important role in suppressing anticancer immune responses [115], can inhibit antitumor activity of T cells by producing immunosuppressive cytokines IL-10 and transforming growth factor-β (TGF-β) [116]. In breast cancer resection model mice, orally administrated naringenin can reduce the outgrowth of metastases by enhancing antitumor activity of T cells and inhibiting immunosuppression caused by regulatory T cells [84]. Besides, the natural flavone tetramethoxyluteolin inhibited the release of human mast cell inflammatory mediators by decreasing intracellular calcium levels and NF-κB activation at both the transcriptional and translational levels, thus suppressing inflammatory conditions [85].

In summary, daily intake of 15–25 mg quercetin or flavonols have a negative relationship with lung cancer and breast cancer in case-control studies. In vitro, dietary flavonoids induce apoptosis by increasing apoptotic protein expression, ROS production, intracellular caspase-9, caspase-3, Apaf-1, and caspase-3 level, and decreasing AKT expression, inhibiting autography, and STAT3 signaling; Flavonoids modulate autography through inhibiting mTOR signaling pathway and proteasome; Flavonoids target NF-κB to inhibit cancer development and inflammatory responses. 

## 4. Preventative Roles of Flavonoids in Neurodegenerative Diseases

The neurodegenerative diseases, including Parkinson’s disease (PD), Alzheimer’s disease (AD), amyotrophic lateral sclerosis (ALS), and Huntingdon’s disease (HD), are a group of heterogeneous disorders due to the progressive loss of neuronal function in a specified area of the brain [117,118]. These disorders are caused by oxidative stress, neuroinflammation, abnormal accumulation of proteins, impaired mitochondrial function, and apoptotic activation in the brain, as well as interaction with lifestyle, genetic, and environmental factors [119,120,121]. Currently, about 50 million people are affected by neurodegenerative diseases [122], which is estimated to increase to 130 million by 2050 [123]. Dementia is the major cause of disability, institutionalization, and mortality, and the global cost is USD 1 trillion [123]. Flavonoids can cross the blood-brain barrier and possess multiple biological properties, so they have the potential to prevent the development and progression of neurodegenerative disease [124,125].

**Table 3 antioxidants-12-00527-t003:** Neuro-protective roles of flavonoids in experimental models.

Flavonoids	Dosage	Duration	Models	Effects	Molecular Mechanisms	Ref.
EGCG and tea polyphenols	2 and 10 mg/kg	14 d	PD mice model	↓Dopaminergic neurodegeneration	Antioxidative and iron-chelating properties	[125]
Quercetin	50 mg/kg	14 d	PD mouse model	Antiparkinsonian properties	↑AchE and antioxidant activities	[126]
Anthocyanins	20 mg/kg	84 d	Transgenic R6/1 HD male mice	↑Spatial cognition learning ability	↓Oxidative status	[127]
Quercetin	25 mg/kg	90 d	Aged triple transgenic AD mice model	↓Alzheimer’s disease pathology ↑Cognitive and emotional function	-	[128]
Fisetin	0.05%	180 d	AD transgenic Mice model	Maintains cognitive function	Modulation of p25 and inflammatory pathways	[129]
Genistein	40 μM	48 h	Aβ-induced hippocampal cell	↓Aβ-induced neuronal apoptosis	Antioxidative properties Estrogen receptor-mediated pathway	[130]
Morin	1–10 μM	6 h	Human neuroblastoma cells	↓Neuronal apoptosis and tau phosphorylation	↓GSK3β	[131]
7,8-dihydroxyflavone	5 mg/kg	75 d	ASL transgenic mice model	↑Motor performance ↑Lower motor neuronal survival	-	[132]
EGCG	10 mg/kg	91 d	ASL transgenic mouse model	Neuroprotective effects	↓NF-κB and cleaved caspase-3	[133]
Fisetin	9 mg/kg	80 d	ALS transgenic mice model	Antioxidant and Neuroprotective Effects	↑ERK	[134]
Chrysin	50 mg/kg	14 d	HD rat model	↓Mitochondrial dysfunction and striatal apoptosis	↑Bcl-2 gene ↓Bax-Bad genes	[135]
7,8-dihydroxyflavone	5 mg/kg	119 d	Male R6/1 transgenic mice	↓Cognitive and motor deficits	↑PLCγ1 pathway	[136]

Notes: AD, Alzheimer’s disease; ASL, amyotrophic lateral sclerosis; Aβ, amyloid β; AchE, acetylcholinesterase; Bax, Bcl-2-associated X protein; Bad, Bcl-2-associated death promoter; Bcl-2, B-cell lymphoma 2; ERK, extracellular signal-regulated kinases; GSK3β, glycogen synthase kinase 3 β; HD, Huntingdon’s disease; PLCγ1, phospholipase C, γ1; NF-κB, nuclear factor kappa-light-chain-enhancer of activated B cells. ↓, downregulate; ↑, upregulate.

### 4.1. Parkinson’s Disease

As the second prevalent adult neurodegenerative disorder, Parkinson’s disease (PD) is the result of aging and related changes including successive oxidative stress, mitochondrial dysfunction, and protein misfolding and aggregation. PD is characterized by the loss of dopaminergic neurons in the substantia nigra, the formation of Lewy bodies, and the depletion of striatal dopamine. Its symptoms include bradykinesia, resting tremor, gait difficulty, cognitive impairment, and olfactory dysfunction [137,138]. By the time the first symptom appears, striatal dopamine has been decreased by ~80%, and ~60% of the dopaminergic neurons in the substantia nigra have died [139]. Several studies have demonstrated that flavonoids could effectively target the multiple pathogeneses of PD. In an extremely large epidemiological study (about 130,000 people were followed for 20–22 years), male participants who consumed the most flavonoids (including berry fruits, red wine, tea, apples, and orange) had a 40 percent lower risk of developing Parkinson’s disease than those who consumed the least [140]. Ginkgo biloba extract, which is rich in flavonoids, was found to have a protective effect on dopaminergic neurons in PD animal models [141]. Green tea and EGCG were also reported to prevent the loss of dopaminergic neurons in the substantia nigra and depletion of striatal dopamine in mice models of Parkinson’s disease [126]. Besides, quercetin could protect mesencephalic dopamine neurons from injury and attenuate apoptosis induced by oxidative stressors in primary rat mesencephalic cultures [142]. Another study demonstrated that orally administrated quercetin notably improved motor balance and increased antioxidant activities such as Na-K ATPase, SOD, and glutathione peroxidase in the PD mouse model [127]. These results suggest a promising therapy that flavonoids may target multiple pathogeneses to prevent PD before overt disease onset.

### 4.2. Alzheimer’s Disease

Alzheimer’s disease (AD) is the most common neurodegenerative brain disorder and is considered the major cause of death in elderly people. It is characterized clinically by cognitive decline and progressive memory loss, and neuropathologically by the presence of neurofibrillary tangles containing tau and extracellular neuritic plaques mainly due to aggregation of amyloid β (Aβ) [143,144,145]. Natural flavonoids have been identified as a potential therapy for the prevention or treatment of AD. For example, long-term administration with green tea catechins was found to improve learning ability and cognitive function in rats [146]. A study showed that quercetin can improve cognitive and emotional functions in aged triple transgenic AD mice models, and alleviate histological indicators of AD, including aggregation of tauopathy and Aβ, microgliosis, and astrogliosis in the amygdala and hippocampus [128]. Fisetin was reported to maintain cognitive function by modulating p25 and inflammatory pathways [129], and treatment with green tea flavonoids can attenuate Aβ-induced cytotoxicity to the primary prefrontal cortical neurons in rats [147]. The neuroprotective properties of flavonoids on AD have been shown not only in Aβ-induced of neuronal death but also in those of oxidative stress-caused neuronal death. A study showed that genistein can inhibit Aβ-induced hippocampal neuronal apoptosis via the estrogen receptor-mediated pathway at the nanomolar level (100 nM) [130]. Ginkgo biloba extract rich in flavonoids can ameliorate oxidative stress or Aβ peptides-induced toxicities in hippocampal cells [148]. Rutin can alleviate oxidative stress and suppress the formation of malondialdehyde and glutathione disulfide in SH-SY5Y cells [149]. Furthermore, morin, a natural flavonoid existing in onion and apple, was reported to inhibit cell apoptosis and GSK3β-mediated tau phosphorylation both in human neuroblastoma cells and in the mouse model of AD [131]. These studies indicated that flavonoids are identified as potential therapy for the prevention or treatment of AD.

### 4.3. Amyotrophic Lateral Sclerosis

Amyotrophic lateral sclerosis (ALS) is a fatal neurodegenerative disease characterized by the degeneration of cortical motor neurons and anterior horn cells, along with muscle weakness and paralysis [150]. Approximately 10% of ALS cases are caused by heritable gene mutations in the copper/zinc SOD 1 gene, while the majority of cases are sporadic [151]. Given the fact that ALS is a multifactorial disease, bioactive compound flavonoids that target multiple sites are needed to prevent this devastating neuron disease. For example, 7,8-DHF supplementation at one month of age could protect against the age-dependent impairment of motor performance in the SOD1-G93A-induced ASL mouse model [132]. Oral administration of EGCG was reported to delay symptoms onset and prolong lifespan in SOD-G93A-treated mice, as well as promoting motor neuron survival and reducing inflammatory markers [133]. Another study has shown that flavonol fisetin protected neuron cells from ROS damage and improved the pathological behaviors in mutant hSOD1 ALS mice models via ERK activation [134]. Although some progress has been made, more insight into ALS prevention or treatment using flavonoids remains to be explored in the future.

### 4.4. Targeting NF-κB

Huntingdon’s disease (HD) is a genetic autosomal disease resulting from the amplification of cytosine adenine guanine trinucleotide repeats of the Huntington gene [152]. The clinical symptoms of HD include progressive, involuntary movements, psychiatric disturbances, dementia, choreoathetosis, and premature death. The pathological characteristic is mainly the selective degeneration of striatal neurons that produce g-aminobutyric acid in the deep layer of the cerebral cortex [153]. With ongoing research for HD prevention, several flavonoids have been considered to prevent this disease. For example, oral administration with chrysin can promote the survival of striatal neurons, as well as inhibit cell death and oxidative stress in the 3-NP-induced HD rat model [135]. Besides, chronic administration of 7,8-DHF to the HD mouse model can reduce inflammation and the loss of striatal volume, promote neurotrophic factor signaling, and delay motor and cognitive deficits [136]. Daily injection of kaempferol and genistein can prevent the loss of striatal neurons, attenuate motor deficits, and decrease nerve cell death [127,154]. Besides, quercetin, hesperidin, and naringin were found to prevent cognitive impairment in the 3-nitropropionic acid-treated HD mouse model [155,156,157].

In summary, flavonoids prevent multiple neurogenerative diseases via reducing dopaminergic neurodegeneration, Aβ-induced neuronal apoptosis, neuronal apoptosis and tau phosphorylation, mitochondrial dysfunction and striatal apoptosis, cognitive and motor deficits, motor dysfunction phenotype and promoting spatial cognition learning ability, cognitive and emotional function, and motor neuronal survival. The molecular mechanisms mainly include modulation of NF-κB and cleaved caspase-3, Bcl-2 gene, Bax-Bad genes, PLCγ1 pathway, and inflammatory and oxidative signaling pathways.

## 5. Discussion and Conclusions

Flavonoids are beneficial bioactive molecules which present in human foods. Emerging evidence is elucidating the mechanisms by which bioactive food components influence health. Here, we reviewed how flavonoids show strong anti-inflammatory and antioxidative properties. Additionally, we discuss the outcomes with significant differences in dosage and duration of flavonoid treatments.

Studies suggest that flavonoids have a pronounced effects on obesity by suppressing adipogenesis, promoting energy expenditure, and alleviating insulin resistance. The detailed data is provided in Table 1. Flavonoids have the ability to modulate almost all of the known obesogenic pathways. The effective dosage of anti-obesity is 20–50 μM in vitro and 1–200 mg/kg BW in vivo. In 3T3-L1 cell models, flavonoids including fisetin, apigenin, and anthocyanins decrease adipogenesis by downregulating PPAR-γ expression at a dose of 10–50 μM in a concentration-dependent manner [10,25,26]. The most effective treated duration of fisetin, apigenin, and anthocyanins is about 12–24 h, for they mainly suppress the early stage of preadipocyte differentiation [10,25,26]. However, isoflavones such as genistein and daidzein inhibit adipogenic differentiation of AD-MSCs from the mid-phase stage, of which the effective treated duration is 12–24 d [27]. The significant differences in duration of treatment may be attributed to the participation of complex pathways in normal adipocyte differentiation. PPAR-γ is one of the major transcription factors that co-ordinate the expression of genes which create and maintain the adipocyte structure in the initial phase of adipogenesis [158,159,160]. Fisetin, apigenin, and anthocyanins target PPAR-γ and thus suppress the early stage of preadipocyte differentiation. Moreover, in obese animal models, the dosage of hesperidin and naringin to regulate glycogen synthesis and gluconeogenesis is 200 times more than that of myricetin and genistein to regulate IR/IRS-1 and PI3K/Akt signaling [32,33,34,35]. In the liver, the defective PI3K/Akt pathway promotes gluconeogenesis by increasing the expression of PEPCK and G6P and inhibits glycogen synthesis by suppressing GK and GSK [56,58]. The significant differences in dose of treatment may be attributed to glycogen synthesis/gluconeogenesis being downstream of/IRS-1 and PI3K/Akt signaling. Although flavonoids have demonstrated remarkable anti-obesity properties, several issues require further investigation, especially the bioavailability. The doses investigated do not necessarily reflect the concentrations achieved by anormal non-supplemented human diet.

Flavonoids exert anti-cancer roles via apoptosis induction, autophagy network, and NF-κB regulation. The detailed data is provided in Table 2. From these data, we observed that flavonoids, including quercetin, naringenin, apigenin, and anthocyanins were those presenting more promising results as they were able to significantly modulate different anti-obesity mechanisms, both in vitro and in vivo. The effective dosage of anti-cancer is 10–300 μM in vitro and 15–25 mg/day in vivo. In case-control studies, daily intake of 15–25 mg (about 50–82 μmol) quercetin can protect against lung cancer and reduce bioactivation of carcinogens [72,73]. In vitro, the effective dose of quercetin that inhibits breast cancer cells is 20–100 μmol/L [77,79]. The differences between doses of in vivo and in vitro may be attributed to the absorptivity and bioavailability of quercetin. Dietary quercetin is absorbed by intestinal epithelium cells and deployed into circulation, which subsequently enter target tissues and cells, where the concentration of quercetin is far less than dietary doses. Moreover, the effective experimental dosage of flavonoids is about 10–120 μM, while the dosage of hesperetin is 100–300 μM. Hesperetin inhibited the cell viability of esophageal cancer cell lines in a concentration-dependent and time-dependent manner [76]. In vivo, hesperetin also inhibited tumor growth in a dose-dependent manner and induced obvious cell apoptosis in the tumor mass [76]. These differences may be attributed to different antioxidative properties of flavonoids. However, large scale clinical studies would be required for the clinical applications of these flavonoids.

Natural flavonoids exhibit plenty of beneficial anti-neuroinflammatory effects and prevent multiple neurogenerative diseases through regulating oxidative stress, neuroinflammation, accumulation of proteins, and mitochondrial function, and activating apoptosis. The detailed data is provided in Table 3. The effective dosage of anti-neuroinflammation is 10–40 μM in vitro and 2–200 mg/kg BW in vivo. There exists significant difference in duration of treatment. In drug-induced mice models, the duration of flavonoids to exert neuroprotective effects is only 14 d [126,127,135]. However, in transgenic mice models, the duration is up to 84–180 d [127,128,129,130,131,132,133,134,136]. The significant differences in duration of treatment may be due to the difference in time of flavonoids treatment. In drug-induced mice models, mice were orally administered with flavonoids for 10 days and received a drug induction following 4 days, which demonstrated that intake of flavonoids before neurogenerative diseases occur could protects cognitive and emotional function and alleviate oxidative stress. In transgenic mice models, flavonoids alleviate several pathological behaviors of neurogenerative diseases and protect cells from ROS damage, which indicate that flavonoids could also treat neuroinflammation after neurogenerative diseases occur, but it takes a longer time. Remarkably, in the HD rat model, rats administered with chrysin after drug induction and chrysin at 50 mg/kg BW, orally, for 14 days, showed improved behavioral performances and regulation of complex activities in mitochondria, which showed that chrysin may have significant neuroprotective effect in 3-NP induced neurotoxicity. From the summary of current research, different types of natural flavonoids share similar anti-neuroinflammatory mechanisms without obvious difference, among which, inflammatory and oxidative signaling pathways have been widely studied, while other pathways are less studied and need further study.

Overall, flavonoids exert anti-obesity and anti-neuroinflammatory effects at a dosage of 10–50 μM in vitro, but inhibit cancer cells at a much higher dosage, up to 10–300 μM. In vivo, the effective duration of flavonoids to prevent obesity, cancer, or neuroinflammation is 2–35 days, but for alleviating neurogenerative diseases, it takes a longer time, about 80–180 days. These data help us to further investigate natural flavonoids and offer ideas for finding new dietary supplements, which might guide healthy dietary habits for people of different physical status.

## Figures and Tables

**Figure 1 antioxidants-12-00527-f001:**
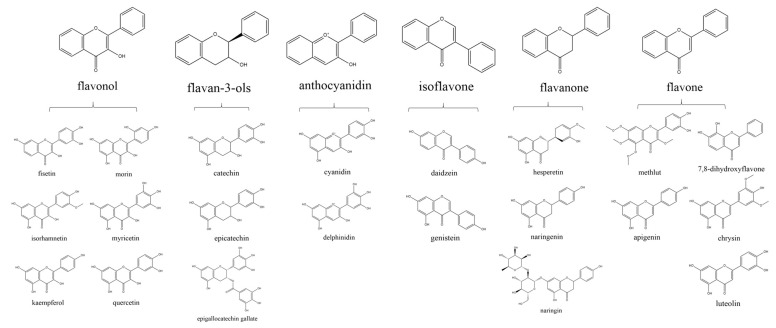
Classification and chemical structures of flavonoids. The chemical structures of typical flavonols (fisetin, isorhamnetin, kaempferol, morin, myricetin, and quercetin), flavan-3-ols (catechin, epicatechin, and epigallocatechin gallate); anthocyanidins (cyanidin and delphinidin), isoflavones (daidzein and genistein), flavanones (hesperetin, naringenin, and naringin), and flavones (methlut, apigenin, luteolin, 7,8-dihydroxyflavone, and chrysin) are shown.

**Figure 2 antioxidants-12-00527-f002:**
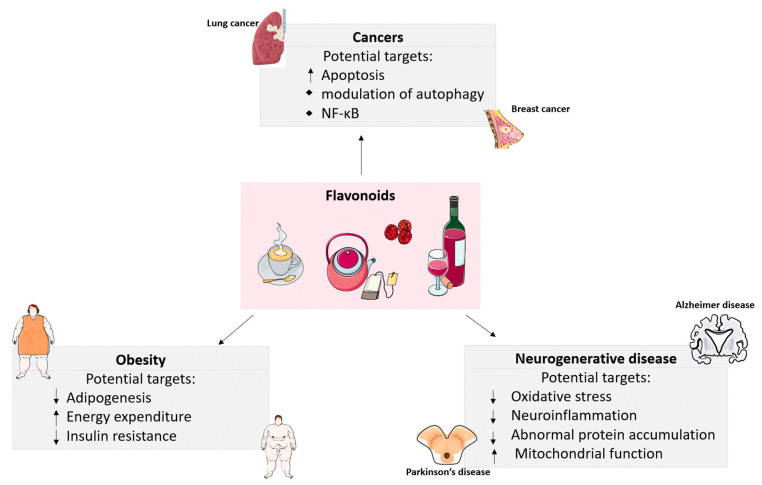
Potential mechanisms of flavonoids intervening in obesity, cancers, and neurogenerative diseases. Flavonoids can control body weight by suppressing adipogenesis, promoting energy expenditure, and alleviating insulin resistance; Flavonoids exert anti-cancer roles via apoptosis induction, autophagy network, and NF-κB regulation; Flavonoids prevent neurogenerative diseases via alleviating oxidative stress, neuroinflammation, protein accumulation, and increasing mitochondrial function. ↓, downregulate; ↑, upregulate.

**Figure 3 antioxidants-12-00527-f003:**
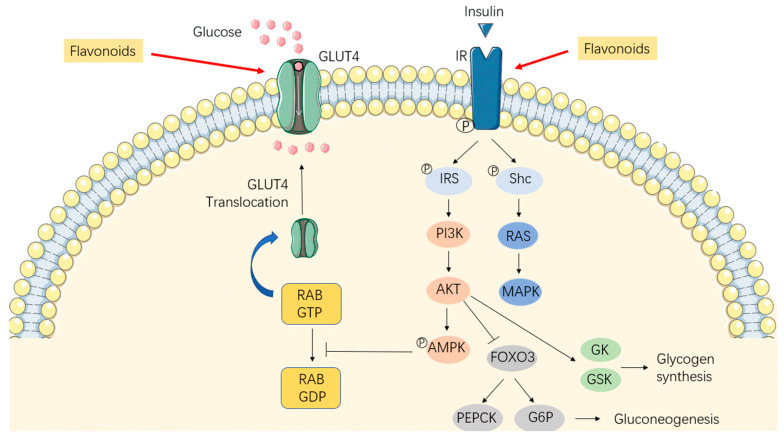
The potential mechanisms of how flavonoids intervene in insulin sensitivity. Flavonoids restore the phosphorylation of IRS and the activation of the PI3K/Akt pathway, which promotes the phosphorylation of AMPK and translocation of GLUT4 to the cell membrane, contributing to increased glucose uptake in the skeletal muscle and adipose tissue. In the liver, flavonoids inhibit gluconeogenesis and promote glycogen synthesis by improving the PI3K/Akt pathway. AMPK, AMP-activated protein kinase; AKT, protein kinase B; FOXO3, Forkhead box O3; GDP, guanosine diphosphate; GK, glucokinase; GLUT4, glucose transporter type 4; GSK, glycogen synthesis kinase; GTP, guanosine triphosphate; G6P, glucose 6-phosphate; IR, insulin receptor; IRS, insulin receptor substrate; MAPK, Mitogen-activated protein kinase; PI3K, phosphatidylinositol 3-kinases; PEPCK, phosphoenolpyruvate carboxykinase.

**Figure 4 antioxidants-12-00527-f004:**
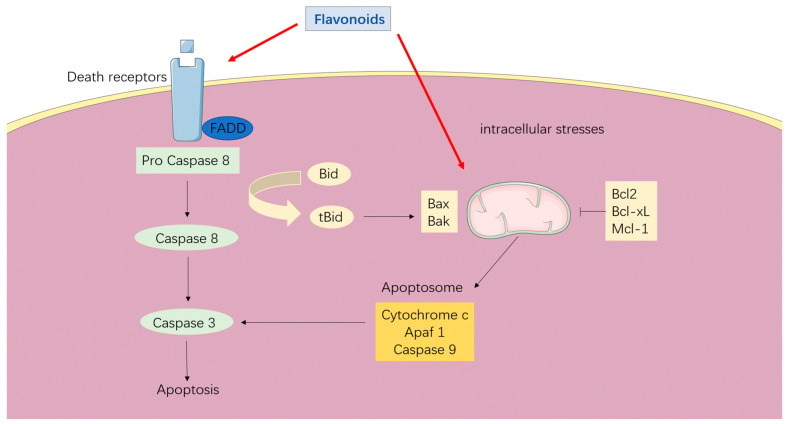
Flavonoids promote intrinsic and extrinsic apoptosis pathways. Two principal pathways are involved in cell apoptosis. The intrinsic pathway is triggered by diverse intracellular stresses, which activate cytochrome c, Apaf-1, and caspase-9 to form an apoptosome through regulation of BH3-only and Bcl2-like proteins and promote apoptosis. The extrinsic pathway induced when the TNF family binds to death receptors on the cell surface, which activates caspase-8 and caspase-3 through Fas-associated death domain protein (FADD) and provokes cellular destruction. Flavonoids induce cell apoptosis via intrinsic and extrinsic pathways.

**Table 1 antioxidants-12-00527-t001:** Effects of flavonoids on obesity in experimental models.

Flavonoids	Dosage	Duration	Models	Effects	Mechanisms	Ref.
Anthocyanins	50 μg/mL	1–3 d	3T3-L1 cells	Anti-adipogenesis	↓PPAR-γ expression	[25]
Fisetin	25 μM	0–2 d	3T3-L1 cells	Anti-adipogenesis	↑SIRT1 expression ↓PPAR-γ	[26]
Genistein	20 μM	12–42 d	AD-MSCs	Anti-adipogenesis	↑Wnt/β-catenin signaling	[27]
Daidzein	20 μM	12–24 d	AD-MSCs	Anti-adipogenesis	↑Wnt/β-catenin signaling	[27]
Apigenin	50 μM	2 d	3T3-L1 cells	Anti- adipogenesis	↓Adipogenic gene ↓Cell cycle	[10]
Luteolin	0.01%	84 d	High-fat-fed mice	↑Browning and thermogenesis	↑AMPK/PGC1α pathway	[28]
Flavan-3-ols	10 mg/kg BW	20 h	AR blocker-treated mice	↑Energy expenditure	↑Sympathetic nerve	[29]
Epigallocatechin-3-gallate	0.2%	32 d	High-fat-fed mice	↑Thermogenesis and mitochondrial biogenesis	↑Mitochondrial DNA replication and AMPK activation	[30]
Epigallocatechin gallate	1.0%	4–7 d	High-fat-fed mice	↑Energy excretion	↓Food digestibility ↑Fat oxidation	[31]
Genistein	1 mg/kg BW	45 d	High-fat–high-fructose-fed mice	↑Insulin sensitivity	↑IRS phosphorylation ↑PI3K/Akt pathway	[32]
Myricetin	1 mg/kg BW	14 d	High-fructose-fed mice	↓Insulin resistance	↑IR and IRS1 phosphorylation ↑PI3K/Akt signaling	[33]
Myricetin	1 mg/kg BW	14 d	High-fructose-fed mice	↓Insulin resistance	Binding µ-opioid receptor ↑IRS-1, PI3K, and Akt	[34]
Hesperidin	0.2 g/kg BW	35 d	Type-2 diabetic mice	↑Glycogen synthesis	↑GK	[35]
Naringin	0.2 g/kg BW	35 d	Type-2 diabetic mice	↓Gluconeogenesis	↓PEPCK and G6P	[35]
procyanidins	80 mg/kg BW	35 d	High-fat-fed mice	↓Insulin resistance	↑GK and hepatic glycogen concentration	[36]

Notes: AD-MSCs, adipose tissue-derived mesenchymal stem cells; BW, body weight; AMPK/PGC1α, AMP-activated protein kinase/peroxisome proliferator-activated receptor-gamma coactivator-1α; GK, glucose kinase; G6P, glucose-6-phosphatase; IRS, insulin receptor substrate; PEPCK, phosphoenolpyruvate carboxykinase; PI3K/Akt, phosphatidylinositol 3-kinases/protein kinase B; PPAR-γ, peroxisome proliferator-activated receptor-gamma; SIRT1, sirtuin1; Wnt/β-catenin, the wingless-type MMTV integration site family/canonical. ↓, downregulate; ↑, upregulate.

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
