# Peer review of "Evidence of Flavonoids on Disease Prevention"

_antioxidants, 2023, doi:10.3390/antiox12020527_

Round 1

Reviewer 1 Report

In this manuscript, authors provided the summary of the experimental results concerning the effects of flavonoids consumption on obesity, cancers, and neurogenerative diseases and tried to discuss the molecular mechanisms of flavonoids preventing these diseases. The authors attempted to review and discuss the findings in the field of flavonoids’ research, including the effects and molecular mechanisms of different flavonoids on obesity, cancer, and neurodegenerative disease prevention. In fact, since there have been many excellent review papers on the efficacy of flavonoids so far, in this review paper, it would be better to try to present something different from other existing review papers, by providing the more detailed experimental information including the difference of dosages and durations, etc. of flavonoids. The authors are recommended to provide information on more detailed experimental conditions, including treatment concentrations and treatment times of flavonoids used in the experiments of many references introduced in this review paper, and to encourage more in-depth discussion of these results.

Author Response

Thank you very much. Your suggestion is of great help to our manuscript. In table 1-3, we added more detailed experimental conditions. In addition, we try more in-depth discussion in our article. We outline these changes with red in the re-submitted manuscript. Thank you again for your comments!

Reviewer 2 Report

The research topic is very interesting and topical.

The considerable amount of scientific articles on the subject proves it, therefore a well-done review can be of help to anyone approaching this topic. The manuscript is well organized and reads fluently. Comprehension is facilitated by schematic figures and exhaustive tables. Some small typewriting errors (i.e. Tab. 1, first line: "tin"... it is intuitive that something is missing!), can be easily corrected while correcting the proofs.

Author Response

Thank you very much. We have checked and revised the manuscript according to the comments and outline these changes with red in the re-submitted manuscript. Thank you again for your comments!

Reviewer 3 Report

This broad review paper summarizes several aspects of flavonoids in their potential role in preventing multiple chronic diseases. Diagrams and tables are helpful, although not all categories have such helpful visualization (e.g., insulin resistance). The review seems to only provide one side, with no limitations or knowledge gaps identified, nor are any recommendations made for next step and future studies. The conclusion is abrupt and short, when a deeper discussion is warranted given the broad-sweeping nature of the review. Also, the last few sentences seem to overstate the known data by jumping to the idea of developing therapeutics without providing any substantive discussion and data that would support that. Tables also need to be proofread for consistency and accuracy, for example Table 1 lists the compound as "tin" and Table 2 describes a study as "10054 participants in Finnish" when the other studies are described more effectively (e.g., case-control).

Author Response

Response: Thank you very much. We have revised Table1 and 2 according to the comments. In addition, we more in-depth discussion and revised the conclusion. We outline these changes with red in the re-submitted manuscript. Thank you again for your comments!

Round 2

Reviewer 1 Report

In this revised manuscript, authors provided the duration information in the tables. The authors are recommended to provide more in-depth summary and discussion, based on the dosage (after converting to the same unit) and duration information. An in-depth summary and discussion of the results with significant differences in the concentration and duration treated should be provided. I believe that providing an in-depth summary and discussion of outcomes with significant differences in concentration and duration of treatment would result in a higher quality review paper.

Author Response

Dear reviewer, 

Thank you very much. Your suggestion is really of great help to our manuscript. We try to provide an in-depth discussion of outcomes with significant differences in concentration and duration of treatment in our article. We outline these changes with red in the re-submitted manuscript. If we mis-understand your comments, please let me know and we will try our best to revise it. Thank you again for your comments!

Round 3

Reviewer 1 Report

In this revised manuscript, authors try to provide some discussion on the outcomes with significant differences in concentration and duration of treatments in the article. However, the authors are still required to provide more in-depth summary and discussion, based on the dosage and duration information. Authors are recommended to check the each experiment in the reference articles and in-depth summary and discussion of the results with significant differences in the concentration and duration treated should be provided. I believe that providing an in-depth summary and discussion of outcomes with significant differences in concentration and duration of treatment would result in a higher quality review paper.

Author Response

Dear reviewer,

Thank you very much for your suggestion. We provide more in-depth discussion based on the dosage and duration information. We outline these changes with red in the re-submitted manuscript. Thank you again for your comments!
